# V-CARE (Virtual Care After REsuscitation): Protocol for a Randomized Feasibility Study of a Virtual Psychoeducational Intervention After Cardiac Arrest—A STEPCARE Sub-Study

**DOI:** 10.3390/jcm14134429

**Published:** 2025-06-22

**Authors:** Marco Mion, Gisela Lilja, Mattias Bohm, Erik Blennow Nordström, Dorit Töniste, Katarina Heimburg, Paul Swindell, Josef Dankiewicz, Markus B. Skrifvars, Niklas Nielsen, Janus C. Jakobsen, Judith White, Matt P. Wise, Nikos Gorgoraptis, Meadbh Keenan, Philip Hopkins, Nilesh Pareek, Maria Maccaroni, Thomas R. Keeble

**Affiliations:** 1Essex Cardiothoracic Centre, Mid and South Essex NHS Foundation Trust, Basildon SS16 5NL, UK; maria.maccaroni@nhs.net (M.M.); thomas.keeble2@nhs.net (T.R.K.); 2Anglia Ruskin School of Medicine and MTRC, Chelmsford CM1 1SQ, UK; 3Neurology, Department of Clinical Sciences Lund, Lund University, SE-221 00 Lund, Sweden; gisela.lilja@med.lu.se (G.L.); mattias.bohm@med.lu.se (M.B.); erik.blennow_nordstrom@med.lu.se (E.B.N.); dorit.toniste@med.lu.se (D.T.); katarina.heimburg@med.lu.se (K.H.); 4Department of Neurology, Skåne University Hospital, SE-221 85 Lund, Sweden; 5Department of Rehabilitation Medicine, Skåne University Hospital, SE-221 85 Lund, Sweden; 6Department of Cardiology, Skåne University Hospital, SE-221 85 Lund, Sweden; 7Sudden Cardiac Arrest UK, Benfleet SS7 5HA, UK; paul.swindell@scauk.org; 8Department of Clinical Sciences Lund, Section of Cardiology, Skåne University Hospital, SE-221 85 Lund, Sweden; 9Department of Anaesthesia and Intensive Care, Helsinki University Hospital and University of Helsinki, 00029 Helsinki, Finland; markus.skrifvars@hus.fi; 10Anesthesiology and Intensive Care, Department of Clinical Sciences, Helsingborg Hospital Lund, Lund University, SE-221 00 Lund, Sweden; niklas.nielsen@med.lu.se; 11Anesthesia and Intensive Care, Department of Clinical Sciences Lund, Lund University, SE-221 00 Lund, Sweden; 12Copenhagen Trial Unit, Centre for Clinical Intervention Research, The Capital Region, Copenhagen University Hospital—Rigshospitalet, 2100 Copenhagen, Denmark; 13Department of Regional Health Research, The Faculty of Health Sciences, University of Southern Denmark, 5230 Odense, Denmark; 14CEDAR (Centre for Healthcare Evaluation, Device Assessment, and Research), Cardiff and Vale University Health Board, Cardiff CF14 4XN, UK; judith.white3@wales.nhs.uk; 15General Adult Intensive Care Unit Critical Care Directorate, University Hospital of Wales, Heath Park, Cardiff CF14 4XW, UK; mattwise@doctors.org.uk; 16Barts Health NHS Trust, London E1 1BB, UK; nikolaos.gorgoraptis@nhs.net; 17Barts Heart and Thorax Centre, St. Bartholomew’s Hospital, London EC1A 7BE, UK; meadbh.keenan@nhs.net; 18Department of Critical Care, King’s College Hospital NHS Foundation Trust, London SE5 9RS, UK; p.hopkins@nhs.net; 19Department of Cardiology, King’s College Hospital, NHS Foundation Trust, London SE5 9RS, UK; nileshpareek@nhs.net; 20School of Cardiovascular Metabolic Medicine & Sciences, BHF Centre of Excellence, King’s College London, London SE1 7EH, UK

**Keywords:** out-of-hospital cardiac arrest, psychoeducation, telemedicine, quality of life, family caregivers, survivors

## Abstract

**Background:** Out-of-hospital cardiac arrest (OHCA) survivors and their relatives may face challenges following hospital discharge, relating to mood, cognition, and returning to normal day-to-day activities. Identified research gaps include a lack of knowledge around what type of intervention is needed to best navigate recovery. In this study, we investigate the feasibility and patient acceptability of a new virtual psychoeducational group intervention for OHCA survivors and their relatives and compare it to a control group receiving a digital information booklet. **Methods:** V-CARE is a comparative, single-blind randomized pilot trial including participants at selected sites of the STEPCARE trial, in the United Kingdom and Sweden. Inclusion criteria are a modified Rankin Scale (mRS) ≤ 3 at 30-day follow-up; no diagnosis of dementia; and not experiencing an acute psychiatric episode. One caregiver per patient is invited to participate optionally. The intervention group in V-CARE receives four semi-structured, one-hour-long, psychoeducational sessions delivered remotely via video call by a trained clinician once a week, 2–3 months after hospital discharge. The sessions cover understanding cardiac arrest; coping with fatigue and memory problems; managing low mood and anxiety; and returning to daily life. The control group receives an information booklet focused on fatigue, memory/cognitive problems, mental health, and practical coping strategies. **Results:** Primary: feasibility (number of patients consented) and acceptability (retention rate); secondary: satisfaction with care (Client Satisfaction Questionnaire 8 item), self-management skills (Self-Management Assessment Scale) and, where available, health-related outcomes assessed in the STEPCARE Extended Follow-up sub-study including cognition, fatigue, mood, quality of life, and return to work. **Conclusions:** If preliminary insights from the V-CARE trial suggest the intervention to be feasible and acceptable, the results will be used to design a larger trial aimed at informing future interventions to support OHCA recovery.

## 1. Introduction

Every year more than 300 000 individuals in Europe suffer an out-of-hospital cardiac arrest (OHCA); only around 8% of them survive to hospital discharge [1,2].

Most survivors make a ‘good’ recovery—that is, they achieve a quality of life akin to what they had prior to their OHCA—despite this, difficulties with cognition, mood, and physical recovery have been extensively documented. Cognitive deficits, albeit usually mild, affect as many as 40–50% of survivors 6–12 months after their cardiac arrest [3,4,5]. Impairments are often found in memory, followed by executive functioning and processing speed, although the heterogeneity of the measures and methodology used reduces the generalizability of these results [6,7]. Anxiety and depression have also been described in OHCA survivors, with a recent systematic review reporting symptoms of depression, anxiety and post-traumatic stress disorder (PTSD) in 19%, 26%, and 20% of patients, respectively [8]. Psychological distress has been found to contribute to fatigue severity one month after hospital discharge [9]; high psychological distress can also impact a variety of health behaviours and influence health-related quality of life and risk of cardiovascular disease [10,11]. Rates of fatigue, anxiety, depression, reduced mental function, and disability appear to be the same 1 and 5 years after the event, suggesting limited natural recovery [10]. Returning to daily activities and previous occupational roles is often problematic, with around half of survivors experiencing difficulties or the inability to resume work [12,13,14,15]; these rates remain lower compared to patients recovering from myocardial infarction [16].

Co-survivors, including family members and close friends of survivors, also undergo their own process of recovery. Qualitative studies show that this involves overcoming the initial shock of witnessing the arrest [17] and having to support the survivor in their recovery after discharge from hospital, often with little or no support from healthcare professionals [18]. Persistent symptoms of post-traumatic stress disorder (PTSD) and anxiety have also been documented to be common in this population [19,20,21]; however, no study thus far has assessed the impact of mental interventions to mitigate their psychological distress [22]. Recent research underlines that both survivors and co-survivors have a need for more resources and education on how to best navigate recovery after an OHCA [17,23].

### 1.1. Rationale for a Psychoeducational Intervention

Several questions remain unanswered around the psychosocial impact of survivorship on quality of life, as well as optimal post-discharge rehabilitation and care to maximize recovery for survivors and for their family members. In recent years there has been broad recognition of the importance of providing follow-up for survivors of OHCA and their key supporters (hereby defined as family members, spouses, partners, or friends who provide support from admission to hospital through rehabilitation/recovery and beyond). One such example is a priority-setting exercise completed using the methodology developed by the James Lind Alliance, which identified that the needs of key supporters post-OHCA must be served better, as well as confirming the specific need for post-discharge survivor-focused care [24,25]. Similarly, a statement from the American Heart Association (AHA) also identifies a knowledge gap in cardiac arrest research concerning the role for support networks/groups after cardiac arrest [26]. Current guidelines and position statements all stress the importance of providing appropriate follow-up care, both to survivors and key supporters, whilst highlighting the lack of evidence-based interventions available [26,27,28].

Thus far, few studies have evaluated the impact of post-discharge support and educational interventions on longer-term quality of life. A recent systematic review and meta-analysis highlighted the lack of good-quality evidence and the high heterogeneity in intervention design and the outcome measures used; in addition, rehabilitation interventions have not been described in detail, preventing replication of results [29]. Of the studies published so far, the ‘Activity and Life after Survival of a Cardiac Arrest’ RCT showed significant benefits in the outcomes of cardiac arrest survivors, with additional analyses suggesting a high probability of cost-effectiveness from a societal perspective [30,31]. An intense 11-session individual psychological and education intervention was found to reduce risk of cardiovascular death; however, the impact on quality of life was not measured [32], and the volume and intensity of therapy provided may be a barrier to widespread adoption. A small pilot study focused specifically on chronic fatigue after OHCA showed that an energy-conservation and problem-solving therapy intervention (delivered over the telephone) was feasible, acceptable, and effective in improving outcomes [33]. More recently, a residential rehabilitation intervention focused on fatigue and physical/psychological consequences of an OHCA showed promising results in several domains, such as quality of life, fatigue, and anxiety, even though the high intensity/volume of this intervention may not make it suitable for widespread adoption [34]. A protocol for the remote delivery of individual psychotherapy and cardiac-focused psychoeducation combining mindfulness and exposure-based interventions has also recently been successfully trialled in a small sample of OHCA survivors experiencing PTSD [35]. Other trials focused on exploring interventions to facilitate return to work (ROCK trial [36]), alleviate psychological distress and improve cognitive abilities (ENFORCER trial [37]), and to promote recovery and self-management (CARESSf) [38] are currently ongoing.

The studies currently available suggest that a programme integrating educational elements and skill training (for example, for the management of fatigue) might be appropriate for improving outcomes after an OHCA [35]. To our knowledge, however, an intervention involving both survivors and key supporters, delivered virtually in a group format, has not been trialled yet. The feasibility and acceptability of this model require exploration; a virtual group intervention for elderly people with depression was found to be feasible and effective [39]; however, in another study, a tele-rehabilitation group intervention failed to recruit and retain participants in a geriatric population [40].

### 1.2. Study Aims

The primary aims are to investigate whether the Virtual Care After REsuscitation (V-CARE) intervention—a bespoke psychoeducational, remotely delivered intervention co-designed with cardiac arrest survivors and their key supporters—is feasible and tolerable 2 to 3 months post-discharge from hospital. A secondary aim is to explore whether participating in the V-CARE intervention leads to beneficial effects in self-management skills and higher satisfaction with the treatment received, compared to a control group receiving a digital information booklet only.

## 2. Methods

### 2.1. Study Design

This study, a comparative, single-blind randomized pilot trial investigating the psychoeducational V-CARE intervention, is included as a sub-study in the Sedation, TEmperature, and Pressure after Cardiac Arrest and REsuscitation (STEPCARE) trial, an international, multicenter, randomized, factorial, and superiority trial designed to evaluate optimal post-resuscitation care strategies for unconscious adult patients following OHCA. The study employs a 2 × 2 × 2 factorial design to assess three interventions, namely sedation strategy, temperature management, and mean arterial pressure targets [41]. The trial aims to enrol 3500 participants, with an estimated completion date of 30 June 2026.

The V-CARE study described here includes participants at selected sites of the STEPCARE trial, in the United Kingdom and Sweden, surviving with a good neurological outcome (mRS) ≤ 3. Patients will be recruited after the scheduled 30-day follow-up of STEPCARE.

### 2.2. Patient and Public Involvement

The development of the V-CARE intervention and of the digital information leaflet were informed by an exploratory sequential mixed-method design, engaging both OHCA survivors and their key supporters. This process involved two steps:(1)Running two semi-structured focus groups—one with survivors and one with key supporters—each lasting approximately 90 min, to gather in-depth insights from OHCA survivors and their key supporters regarding the ideal structure, content, and mode of delivery for the proposed psychoeducational intervention.(2)Themes emerging from the focus groups (Appendix A) were used to develop a survey aimed at assessing the generalizability of the opinions expressed. This survey was distributed online via Sudden Cardiac Arrest UK and was available for one month between February and March 2022. A total of 95 responses were collected, comprising 62 survivors and 33 family members (70 women, 25 men). The survey included 26 questions, a mix of closed and open-ended items, to capture detailed feedback on the proposed intervention.

The V-CARE intervention and the digital information booklet were initially drafted by our research group. Feedback was then obtained from survivors and key supporters after draft versions were posted in the private Sudden Cardiac Arrest UK (SCA UK) Facebook group. Both the V-CARE intervention and the digital booklet were then translated into Swedish.

### 2.3. Study Population and Randomization

In addition to the criteria needed for taking part in STEPCARE, OHCA survivors eligible to take part in V-CARE are also required to have an mRS ≤ 3 at the 30-day follow-up and (a) must not have a diagnosis of dementia that would impact their ability to participate and benefit from the sessions and complete baseline/outcome measure (based on a clinician’s evaluation and/or self-report by the patient or family) and (b) must not be actively psychotic/experiencing any other serious acute mental health condition that would affect their participation in this study, based on clinical evaluation.

When recruited, consenting patients will be encouraged to take part in the study together with a ‘key supporter’; however, this will not be a requirement.

Consenting patients will be randomized on an equal basis (1:1) to the V-CARE arm or to the information booklet arm. V-CARE has a randomized block design with a separate randomization list in each country, completed by a local lead site. Custom-built software was developed using Python (v3.12.7) and the Tkinter library to allocate patients into two study conditions—V-CARE or the digital information booklet. A computer-generated pseudo-randomized list with 25 sets of 2 unique numbers per set was pre-generated using an online randomization tool [https://www.randomizer.org/] (accessed on 7 October 2024) to ensure an unbiased allocation process.

### 2.4. Interventions

The V-CARE group intervention programme is described based on the Template for Intervention Description and Replication (TIDieR) checklist (Appendix A).

#### 2.4.1. V-CARE (Virtual CAre After REsuscitation)

The intervention focuses on providing early cardiac-arrest-related education, promoting the development of coping skills, encouraging peer support, and signposting survivors and key supporters to relevant resources/services.

The V-CARE programme involves 4 structured sessions, one a week, each lasting around one hour, supported by PowerPoint presentations (Appendix A). The content focuses on providing education on cardiac arrest and secondary prevention, addressing practical problems (for instance, returning to driving, travelling, living with an implantable cardioverter defibrillator (ICD), etc.) and understanding/coping with fatigue, cognitive changes, and psychological difficulties. As multiple sites in each country are expected to take part in this study, groups will be formed as soon as 3 to 6 patients have been recruited, although smaller groups could be formed if required. Group size will usually be between 4 and 12 participants. The aim is for the intervention to be delivered early to each group, starting around 2 and 3 months after hospital discharge; however, it is expected that it may be delivered to some groups up to—but not beyond—6 months post-discharge.

Each session includes broadly equal time for presentation, questions, and discussion, with the aim of encouraging participants to develop self-management skills. A printout of the PowerPoint slides used in each session is provided in Appendix A.

A single, designated healthcare professional will serve as a facilitator for each group and will be present for all sessions, having been trained by the study team; it is anticipated that it will be delivered by either a clinical psychologist or an occupational therapist, but a clinical nurse specialist could also manage a group. A cardiologist or cardiology resident will attend the first session to answer generic, cardiac-arrest-related questions.

#### 2.4.2. Digital Information Booklet

This intervention, also co-developed with a patient group, consists of an information booklet focused on providing information and coping strategies for fatigue, memory, low mood, anxiety, and cognitive problems after a cardiac arrest. Whereas the content is similar to that covered in the V-CARE arm, these patients are not included in a group and as such do not have access to peer support or to guided discussions and reflections as provided in the group setting. In this intervention arm, tolerability will be assessed by the proportion of participants who complete the post-intervention outcome measures; additionally, participants will be asked to self-report whether they read or used the booklet. A copy of this intervention is provided in the Appendix A.

### 2.5. Outcomes and Outcome Measures

#### 2.5.1. Primary Outcomes

The primary outcomes are the feasibility and tolerability of taking part in a psychoeducational virtual group for OHCA survivors and key supporters.

Feasibility of recruitment will be defined as the number of patients who consented out of all the eligible patients (consent rate).

Tolerability will be measured by the completion rate for those recruited and adherence to the intervention protocol (that is, the number of sessions attended out of the total sessions).

#### 2.5.2. Secondary Outcomes

Satisfaction with care will be assessed by the Client Satisfaction Questionnaire-8 (CSQ-8)—for both patients and key supporters. For self-management the Self-management Assessment Scale (SMASc) will be used—patients only [42]. In the control group, patients will be asked to complete outcome measures around a month after being sent the digital booklet—replicating the month-long length of the V-CARE intervention.

#### 2.5.3. Additional Outcomes

Where available, results of the detailed follow-up sub-study of STEPCARE at 6 months and 12 months will also be used in the analysis of V-CARE; they include measures of cognition (MoCA, SDMT, MFIS IQCODE-CA—patients only), burden (ZBI—key supporters), anxiety and depressive symptoms (HADS—both), post-traumatic stress symptoms (PCL-5—both), health-related quality of life (EQ-5D-5L, WHODAS 2.0—both), life satisfaction (both), and detailed questions about return to work/rehabilitation (patient only) [43].

### 2.6. Data Collection, Statistical Analyses, and Sample Size

For primary outcomes, descriptive statistics will be used to report on recruitment and retention rates. Continuous variables will be presented as the mean (SD) or median (IQR) depending on the distribution of data and percentages for categorical variables.

Using the methodology described by Lewis et al. [44], we set up hypothesis testing around progression criteria that tests against being in the ‘red zone’ based on an alternative of being in the ‘green zone’. Specifically, we aim to investigate that a true feasibility/tolerability outcome is not greater than the upper ‘red limit’ (null hypothesis) vs. a true feasibility/tolerability outcome that is greater than that (alternative hypothesis). Criteria for success and a traffic light system to identify the need for amendment for a fully powered trial are as follows:Feasibility: At least 35% of patients screened as eligible should be recruited into the trial (green), but the trial will not be feasible if recruitment uptake is 20% or less (red). Recruitment of between 20% and 35% may require amendments (amber).Tolerability: The drop-out rate of patients recruited in the trial should be 20% or less (successful completion of the trial is defined as taking part in at least 3 out of 4 sessions of the intervention and complete outcome measures). A drop-out rate of 40% or more will indicate that the full trial is not feasible (red), with a percentage between 20% and 40% suggesting that amendments may be required (amber).Satisfaction with care: No more than 30% of participants in the V-CARE arm should score less than 24/32 on the CSQ-8 (green); if more than 60% of participants score less than 24/32, this will indicate that the trial is not feasible (red), with a percentage between 30% and 60% suggesting that amendments to the protocol may be required (amber) (Figure 1).

A sample size calculation using a one-tailed test with alpha = 0.05 and beta = 0.9 identifies that with reference to the three criteria highlighted in this section, respectively, 78, 131, and 143 participants would be needed to be screened in order to meet the individual power requirement of > 90% (with regard to the second and third criterion, this is based on the assumption of only 35% of screened patients being actually randomized (2), and only half of these patients being in the intervention arm (3)). For this reason, we aim to screen at least 145 participants and recruit at least 50 (25 in each arm). This sample size is considered adequate to estimate key parameters to inform the design of an RCT.

To assess differences between two groups, the analyses will be conducted using independent data, with tests appropriate to the distribution of the data (independent *t*-test or Mann–Whitney U test). Estimated effect sizes (with 95% confidence intervals) for difference will be calculated. All *p*-values will be unadjusted for multiple comparisons due to the exploratory design [45]. Multi-variable logistic regression models will be used to evaluate the associations between disease severity, sociodemographic variables, and outcome measures. Outcomes will be statistically controlled for potentially confounding variables such as age, sex, education, and length of intensive care stay. As this is a feasibility study, the study will be underpowered; therefore, no conclusions on the effectiveness of the intervention will be made. SPSS v 29.0 or above will be used to analyze the data.

### 2.7. Ethics and Consent

This study has been reviewed and approved by ethical committees in the United Kingdom, with the IRAS ID: 32885, and by the Swedish Ethical Review Authority—ID 2024-03579-01. The primary STEPCARE trial, under which this sub-study is conducted, is registered with ClinicalTrials.gov (ID: NCT05564754).

## 3. Conclusions

In this manuscript, we present the protocol for a multi-centre, randomized pilot trial investigating if a remotely delivered psychoeducational intervention for OHCA survivors and their key supporters is feasible, and in addition if there is trend towards beneficial effects compared to a control group receiving a digital information booklet only. We describe the different phases of the study, namely the rationale for this intervention, conceptualization of the project, structure of the intervention, and outcome measure adopted. The primary outcome is the evaluation of the feasibility and tolerability of delivering this intervention—definitions have been provided to measure this outcome. Secondary outcome measures include the Client Satisfaction Questionnaire and the Self-Management Assessment Scale. Criteria for the success of the study have been reported, and a description of the pre-planned statistical analyses provided.

The preliminary results from this pilot trial will provide valuable insights to inform a larger trial to evaluate the effectiveness of an intervention to support recovery post-cardiac arrest. First, the outcome feasibility will provide insights regarding the actual interest of patients and their caregivers to take part in a digital intervention to support recovery in the early phase after a cardiac arrest. Second, the outcomes of tolerability and satisfaction with care will be instrumental for the potential need of adjustments in the design of the intervention for the larger trial. Lastly, hypothesis-generating trends in differences between the two groups in terms of self-management skills and health outcomes will be able to guide the choices of primary outcomes and power calculations for a larger trial.

## Figures and Tables

**Figure 1 jcm-14-04429-f001:**
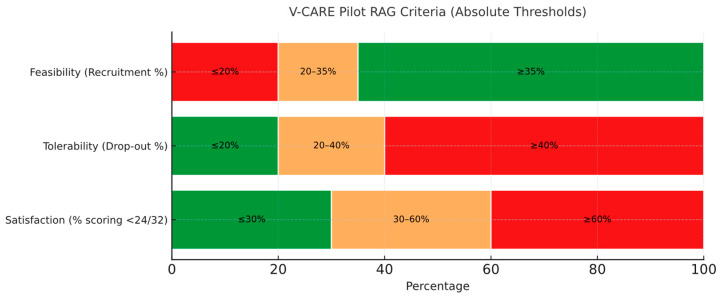
V-CARE pilot feasibility criteria using red–amber–green (RAG) thresholds. The figure presents absolute percentage thresholds across three pilot outcomes: feasibility (percentage of eligible patients recruited), tolerability (percentage of recruited patients dropping out), and satisfaction with care (percentage scoring below 24 out of 32 on the CSQ-8 questionnaire). Thresholds indicate decision-making guidance for progression to a full trial: green = criteria fully met (pilot feasible); amber = criteria partially met (pilot feasible with protocol amendments); and red = criteria not met (pilot not feasible without substantial changes).

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
