# Peer review of "V-CARE (Virtual Care After REsuscitation): Protocol for a Randomized Feasibility Study of a Virtual Psychoeducational Intervention After Cardiac Arrest—A STEPCARE Sub-Study"

_jcm, 2025, doi:10.3390/jcm14134429_

Round 1

Reviewer 1 Report

Comments and Suggestions for Authors

Thank you for the opportunity to review the forthcoming V-CARE (Virtual Care After REsuscitation) protocol – a pilot, comparative, single-blind randomized sub-study of the STEPCARE (Sedation, Temperature, and Pressure after Cardiac Arrest and Resuscitation) trial.

This is a highly relevant and timely topic, addressing an important gap in follow-up care for post-resuscitation patients.

The protocol is logically structured and focuses on evaluating key parameters related to the usability of the educational methods under investigation.

In my opinion, both educational modalities could be useful in the future, depending on the specific characteristics of patients after cardiac arrest.

Therefore, I would suggest including an evaluation of the study cohort in relation to the interventions performed and individual patient characteristics.

Specifically, I recommend taking into account factors such as age, educational background, duration of resuscitation, length of hospital stay, and the presence of post-resuscitation organ dysfunction.

For example, I would expect that patients with prolonged hospitalizations, longer or more complicated resuscitation courses, younger age (with a potentially greater impact on lifestyle), higher education levels, or more serious residual complications (e.g., severe cardiac or multi-organ failure, critical illness myopathy, etc.) might benefit more from personal interaction, the opportunity to ask questions, and more intensive care.

Reviewer 2 Report

Comments and Suggestions for Authors

Dear authors,

Thank you for the opportunity to review your work, which I hope will yield results proportional to the effort invested.

The search for tools that support full recovery after an out-of-hospital cardiac arrest (OHCA) is an ongoing endeavor involving thousands of researchers—not only from a physical standpoint, but also, as in your case, from psychological and functional perspectives, including consideration of family members.

I would like to offer several comments to help improve the clarity and comprehensibility of your manuscript.

Title:
The title is somewhat unclear. It should focus more directly on V-CARE and its objective. Even if this is a sub-study, it has its own merit and constitutes the central topic of the article.

Abstract:
The abstract is appropriate and adequately summarizes the work.

Introduction:
The introduction is rather extensive for an interested reader, providing an overview of various studies on the topic—generally with small patient samples, given the inherent challenges of generating evidence in this field.

Line 98:
Consider simplifying this to: “Rationale for a psychoeducational intervention.”

Objective:
As this is a pilot study, the primary objective should be whether the program can be feasibly implemented or not.

Line 148:
This should be summarized in one sentence: “The primary objective is to investigate whether the V-CARE program is feasible and tolerable at 2 months post-hospital discharge.”
The objective must not be left vague—either it is 2 months, 3 months, or a defined range between 2 and 3 months, but the methodology already describes the design.

Methods:
Both trials are described, but several aspects are missing in the V-CARE description:

Line 170:
Why are only Sweden and the United Kingdom included? Will all patients have mRS ≤3, or only some? If only some, which ones? The selection criteria could introduce bias and should be explained.

Line 190:
As I understand it, the V-CARE intervention and the Digital Information Booklet were reviewed by the research team, but the feedback was obtained from a small group of survivors. Who were they? Why these individuals and not others? Were they more motivated or accessible? Were they volunteers? These details are important for intervention design, as their input has been considered.

Line 196:
I assume psychosis is also assessed by a clinician—please clarify.

Line 229:
If the primary objective is: “…is feasible and tolerable around 2 to 3 months post-discharge from hospital,” we should not later assume a delay until 6 months, unless this is modified in the stated objective. It is understandable that a patient’s psychological state may differ significantly between 2 and 6 months post-discharge.

Line 234:
“The facilitator is a healthcare professional”—Will it always be the same person for each group? Will facilitators receive specific training from the study team? Even if the presentation is standardized, guidance should be provided to ensure consistency across groups, especially since different countries with different cultures are involved.

Line 237:
What happens if there is no cardiologist? Could their presence or absence influence patients’ understanding of their condition and therefore their emotional well-being?

Line 241:
Are the facilitators the same as those for the group intervention? If not, based on what criteria are they selected?

Line 256:
How is tolerability assessed in the Digital Information Booklet group?

The proposed data handling appears appropriate.

The study has obtained approval from the relevant ethics committees.

The biliography is correct.

Conclusion:
Overall, the study is well designed. Although it is to be expected that participants who receive the four group video call sessions may have better outcomes compared to those who only receive the Digital Information Booklet, we must hope that the trial is feasible and the final results.
